# Efficient approximation of neural population structure and correlations with probabilistic circuits

**Koosha Khalvati**
Allen Institute, Seattle, WA 98109
koosha.khalvati@alleninstitute.org

**Samantha N. Johnson**
University of Chicago, Chicago, IL 60637
snjohnso@uchicago.edu

**Stefan Mihalas**
Allen Institute, Seattle, WA 98109
stefanm@alleninstitute.org

**Michael A. Buice**
Allen Institute, Seattle, WA 98109
michaelbu@alleninstitute.org

## ABSTRACT

We present a computationally efficient framework to model a wide range of population structures with high order correlations and a large number of neurons. Our method is based on a special type of Bayesian network that has linear inference time and is founded upon the concept of contextual independence. Our framework is both fast and accurate in approximating neural population structures. Furthermore, our approach enables us to reliably quantify higher order neural correlations. We test our method on simulated neural populations commonly used to generate higher order correlations, as well as on publicly available large-scale neural recordings from the Allen Brain Observatory. Our approach significantly outperforms other models both in terms of statistical measures and alignment with experimental evidence.

## 1 INTRODUCTION

With the rise and fast growth of simultaneous neural population recording, modeling population structures and measuring correlations has become a focus of computational neuroscience (Abbott & Dayan, 1999; Averbeck et al., 2006; Azeredo da Silveira & Rieke, 2021; Urai et al., 2022). Theoretical and Experimental works have demonstrated the necessity of measuring population correlations to investigate information coding (Moreno-Bote et al., 2014; Averbeck et al., 2006), functional connectivity (Dunn et al., 2015), learning (Ganmor et al., 2011), and arousal (Vinck et al., 2015; Doiron et al., 2016). Despite significant progress in recent years, research on measurement and analysis of population correlations still faces significant challenges (Kohn et al., 2016).

Exact measurement of population correlations is an NP-hard problem in the general case since it requires computing every form of dependency among spiking neurons. As a result, researchers have tried to come up with computationally efficient ways of approximation or indirect measurement of neural correlations. Existing approaches are energy-based models rooted in statistical mechanics where the energy function incorporates couplings between subsets of variables (here neurons) (Roudi et al., 2009c; Tkačik et al., 2006; Sohl-Dickstein et al., 2011; Aurell & Ekeberg, 2012). However, these methods often carry auxiliary (and even unrealistic) assumptions about the neural dynamics and do not scale up for large populations (Roudi et al., 2009b).

Notably, generative models commonly used in other domains such as latent variable methods are often not applicable to neural populations as spiking neural data is discrete and sparse (Zhao et al., 2020). Furthermore, various parameters such as behavioural and emotional state of the animal affect firing patterns of neurons even in sensory cortex (Urai et al., 2022). As a result, a recording long enough to train these models contains many external variable changes and confounding factors that make drawing scientific conclusions difficult.

It is important to distinguish whether modeling population correlations is important for predicting joint activity (called *encoding*) vs whether they are a channel for down-stream information flow (call *decoding*) (Pillow et al., 2008). In the encoding paradigm, one needs to know whether it is important to include certain correlations in a generative model of the spiking activity. In the decoding paradigm, one can, e.g., compare a classifier's stimulus prediction accuracy on original and shuffled neural data to assess population correlations (Averbeck et al., 2006; Pillow et al., 2008; Christensen & Pillow, 2022; Runyan et al., 2017). Importantly, the type of used classifier significantly affects the result with no confirmation that the brain utilizes a similar strategy Averbeck et al. (2006).

Here we work in the encoding paradigm and take a probabilistic approach by modeling the joint probability distribution of neural activity with Bayesian networks. Inference is NP-hard in general Bayesian networks (Cooper, 1990), making them impractical to model the population structure. Therefore, we utilize a special family of Bayesian networks with linear inference time, first introduced as "Arithmetic circuits" (Darwiche, 2003; Shen et al., 2016). This family of networks has been designed to take advantage of "context-specific independence" of variables mainly for the purpose of computational efficacy, which also makes it suitable to extract local structures in the data (Boutilier et al., 1996; Shen et al., 2020). We use a modification (and equivalent (Rooshenas & Lowd, 2014)) of arithmetic circuits, Sum-Product Networks (SPNs). SPNs are more known and used by the community (Poon & Domingos, 2011; Sanchez-Cauce et al., 2021).

In particular, we adapt sum-product networks to fit spiking neural data in order to capture a wide range of population correlation/structure from local to global in polynomial time. Due to the efficiency of architecture learning and inference in SPNs, population structure estimation is polynomial in the size of the population. In addition, we suggest a measure of high order population correlations based on our framework. Our results include fitting on neural population simulations constructed with higher order correlations, as well as large-scale neural recording in different brain regions on more than 20 mice. Our framework outperforms both energy-based and latent variable models for neural population structure estimation.

## 2 PROBLEM DEFINITION AND RELATED WORK

One of the critical problems in computational neuroscience is providing an accurate statistical description of spike trains in a population of neurons. As the full representation of the data, i.e. raw spike times, is high dimensional, spike trains are binned into small time windows. The time bin should be short enough so each neuron spikes at most once in each bin (with some amount of tolerance in potentially losing some spikes). In addition, this time bin should be large enough that the assumption of temporal independence of spikes holds. With this time binning strategy, each neuron's activity is a binary variable ($S_i$ for neuron $i$ is equal to 1 if there is a spike in the corresponding bin, otherwise 0) and each time bin represents an i.i.d sample/instance. Therefore, spike trains of $N$ neurons for the duration of $T$ would be represented as a binary matrix $D_{K \times N}$ where $K = T/\Delta t$ in which $\Delta t$ is the bin length (Figure 1, left plot). Consequently, the population activity has a probabilistic representation $P(S_1, \ldots, S_N)$, and the problem turns into modelling this joint distribution, given the data. More specifically, the problem is to find a model $m^*$ from a family of models $M$, and optimize its free parameters $\Theta_m$ so as to satisfy the following:

$$m^*, \theta_m^* = \underset{m \in M, \theta \in \Theta_m}{\arg\max} \frac{1}{K} \sum_{k=1}^{K} \log \left( P(d_{1 \times N}^k | m, \theta) \right) \qquad (1)$$

In the existing approaches, $M$ is set to maximum entropy (Ising) models, which are energy-based methods rooted in statistical mechanics (Roudi et al., 2009c; Schneidman et al., 2006). Since learning maximum entropy models is computationally very expensive, these models are restricted to estimate the statistical properties of the population up to a constant order. However, going beyond second order is not computationally feasible. In fact, building the exact generative model is an intractable problem in the general case even for the second order (pairwise) correlations.

Therefore, even a Pairwise Maximum Entropy (PME) model requires further estimation where more accurate approximation algorithms requires thousands of samples for each pair, making them impossible to be used for large populations of neurons (Roudi et al., 2009c; Tkačik et al., 2006; Sohl-Dickstein et al., 2011; Aurell & Ekeberg, 2012). Moreover, there exist plausible scenarios, such as a dichotomized common input to loosely coupled neurons, in which pairwise correlations are

negligible compared to higher order correlations (Macke et al., 2011b; Amari et al., 2003). In fact, low order correlations do not necessarily capture the population structure even when the structure can be explained by "simple" models (Beretta et al., 2018; de Mulatier et al., 2021). Overall, PME is neither computationally efficient nor accurate in population structure estimation.

To improve the structure estimation, some methods modelled higher order correlations through the energy based approaches, e.g. "k-pairwise" correlations(Tkačik et al., 2014), Restricted Boltzmann Machines (RBMs) on top of pair-wised correlations (semi RBMS (sRBMs)) (Köster et al., 2014), and sparse low-order correlations (Ganmor et al., 2011). Founded upon energy based models, these methods also needs computationally expensive sampling strategies to estimate the partition function. Moreover, to approximate the partition function, these methods often rely on pseudo-likelihoods which needs extensive amount of data points to be a accurate.

Here we use a special type of Bayesian network with linear inference, and consequently fast learning time, to estimate distributions of joint neural activity. Our network approximates these distributions by ignoring correlations based on their effect size, rather than statistical order. Moreover, the normalization process to obtain real probability values is also linear in the size of the network. Thus, it is significantly more successful than energy-based models both in terms of efficiency and accuracy.

## 3  MODEL

Our framework is based on Sum-Product Networks (SPNs). An SPN is a rooted directed acyclic graph representing a joint probability distribution of given variables. This distribution is the result of a hierarchical combination of alternating mixtures (sum nodes) and factorizations (product nodes), with given variables as the leaf nodes of the network (Poon & Domingos, 2011; Sanchez-Cauce et al., 2021). Specifically, each *leaf* node (a node with no children) represents a univariate probability distribution. When the variable is categorical, the leaf node is a variable indicator ($\mathbb{I}(x)$). The *scope* of each leaf node is a singleton where its element is the variable that the leaf represents. Other nodes are either *sum* or *product*. A *product* node represents the product of its children (connected by a link). A *sum* node represents the (normalized) weighted sum of its children. The weight of each child is shown as the label of the link from the sum node to that child. The scope of sum and product nodes is the union of scopes of their children. Starting from a leaf as a one-node SPN, sum product networks can be built bottom-up by combining smaller SPNs through sum or product nodes. The root of the SPN represents a valid joint probability distribution if the SPN is *complete* and *decomposable*. An SPN is complete *iff* all children of each sum node have the same scope. A sum-product network is decomposable *iff* all children of each product node have disjoint scopes.

To model the joint probability distribution of the given variables, the weights on the links of sum nodes should be learned with a learning algorithm such as gradient descent or Expectation-Maximization (EM), given the data (Sanchez-Cauce et al., 2021). Similar to other Bayesian networks, the structure of the graph needs to be determined beforehand. One approach is to use a random dense graph and rely on the parameter learning algorithm to select important components. This strategy requires data sets orders of magnitudes larger than that gathered in neural experiments. Network structure can also be learned from the data directly based on the general computational properties of sum and product nodes (Gens & Pedro, 2013). Sum nodes represent the sum of probability distributions over the same set of variables (see *completeness* above). Therefore, they explain the data best if their child distributions represent disjoint states, or in other words different "contexts" in the data. A Product node computes the joint distribution of its children with a product, explicitly carrying an assumption of independence among the factors contributing to the product. Children of a product node can be determined by using a dependence/correlation measure such as mutual information as a link/edge between each pair of features, subjected to a threshold. In other words, to make a tree/sub-tree with a product node as the root, a graph (totally different from the main SPN tree) is built where two feature nodes are connected if and only if they are considered dependent according to a criterion. Each connected component of this graph is a sub-tree in the sum-product architecture where its root is linked to the root product node.

Picturing the data in matrix form where each row is an instance (or sample) and each column represents a feature (or variable), one can think of the action of a sum node as splitting the rows (samples) and that of a product node as splitting the columns (features). This means that recursive splittings of data rows (e.g. by a common clustering method) producing a sum node, and based on

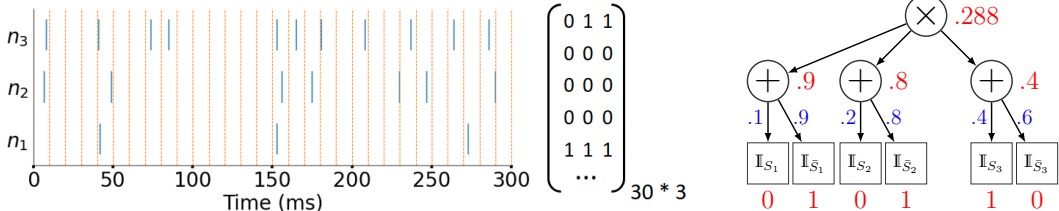

Figure 1: **Modelling Spiking activity with Sum Product Network(SPN)** Left: Neural population activity is often represented by a binary matrix, obtained from binning each neuron's spike train (here 10 ms). Here, the firing rates of neurons 1 to 3 are 10, 20, and 40 Hz. Due to the high correlation of these neurons, bins with all 1, or all 0 are more frequent compared to the activity of 3 independent neurons with the same firing rates. Right: An spn tree (here $nSPN^1$) fit to the population in the left plot. Since it assumes independence, its weights reflect the mean probability of a spike in each time bin after training (shown in blue). Given a data instance (here 001), each node represents the probability of its children, meaning that the root represents the probability (likelihood) of the instance.

variables (e.g. via performing independence tests) resulting in a product node would give us a suitable SPN structure for the given data (Gens & Pedro, 2013; Vergari et al., 2015).

### 3.1 NEURAL POPULATION ANALYSIS WITH *nSPN*S

We construct a family of SPNs that we call Neural SPNs (*nSPN*) to model the joint probability distribution of spiking in populations of neurons. These models are represented by directed graphs where leaves of an *nSPN* representing spiking activity of $N$ neurons correspond to $\mathbb{I}_{s_i}$ and $\mathbb{I}_{\bar{s}_i}$ ($1 \leq i \leq N$) indicating whether neuron $i$ fired or not in a given time bin (see section 2).

**Shallow architecture of *nSPN*s:** The simplest structure of an $nSPN$ consists of a product node as a root, linked to $N$ sum nodes. Each sum node represents the spike probability of a neuron $i$, linked to $\mathbb{I}_{s_i}$ and $\mathbb{I}_{\bar{s}_i}$ leaf nodes. This model, shown as $nSPN^1$ is basically a Naive Bayes model, assuming complete independence among neurons. The right plot in figure 1 shows an example $nSPN^1$ trained on the joint activity of three neurons in the left plot. As mentioned before, each node of an SPN represents the probability of the leaves of the (sub-)tree of which it is the root. As a result, (normalized) link weights of each sum node are aligned with the mean spike probability of the related neuron (note the assumption of independence). Furthermore, given an assignment of probabilities to the leaf nodes and a configuration of spiking activity, the root represents the probability of that configuration's occurrence according to the model (shown by the red number near the root node).

Within the SPN framework, a simple extension of the Naive Bayes model involves the introduction of multiple *contexts*, with population spiking being described by a Naive Bayes assumption *conditioned on the context*. If there are two such contexts with associated probability, the model consists of a sum node as the root linked to two product nodes, each of which a root of an $nSPN^1$ (sub)-tree. Similarly, a model can include an arbitrary number of contexts by increasing the number of product nodes under the root, e.g. $nSPN^b$ for $b$ product nodes/contexts.

In principle, one can construct a shallow graph sufficient to model any data set by adding enough product nodes. One problem with this approach is that there exist simple networks with deeper structure for which a 3-layer model like we have described will require an exponential number of nodes (Delalleau & Bengio, 2011), which in addition to tractability also raises the issue of data limitations. Moreover, all existing parameter learning algorithms for SPNs find local optima and shallow networks get stuck in local optima more frequently. Most importantly, when the data is limited deeper architectures of our framework avoid over-fitting as their structure implements regularization.

**Deep architecture of *nSPN*s:** As computing all correlations is intractable, we seek an approach for constructing graphs that trades some fidelity for efficiency. SPN structure learning algorithms naturally achieve that by separating low correlation variables (according to an independence test and threshold on its $p$-value) at a product node. Importantly, in deep layers of an SPN, variables are considered independent in a given context. We utilized *learnSPN* (Vergari et al., 2015; Molina et al., 2019), shown by $SPN^l$, due to its more conservative approach for splitting. This method has two

hyper-parameters, an independence threshold and a minimum number of data instances to continue splitting. These hyper parameters can be tuned either by a cross validation procedure or based on the data size. Notably, due to our small data size, we applied a Laplace-like regularization to the method to prevent pair-weights of [0, 1] (absolute zero makes the log-likelihood -inf, so we regularized it to $[10^{-5}, 1 - 10^{-5}]$.)

**Measure of Neural Correlation:** The simplest architecture, $nSPN^1$, assumes independence between all neurons. On the other hand, the more complicated structures in our framework only assume independence within each *context* where the number of contexts are always more than 1. This means that in stationary periods where the firing patterns of neurons do not change significantly, $nSPN^1$ fits to the data as well as more complex structures if and only if all neurons are independent of each other. Any improvement in goodness of fit for more complicated structures is due to the existence of neural correlations. As a result, we can use the difference between the average log-likelihood (equation 1) of the best fitting $nSPN$ and $nSPN^1$ as a measure for neural correlations. For example, if our best model is $nSPN^*$, our measure for population neural correlation is $\Delta ll_{SPN} = ll_{nSPN^*} - ll_{nSPN^1}$.

## 4 RESULTS

We start with a simulation of population activity in which we can control the presence of fluctuations. We use a network of homogeneous exponential integrate-and-fire (EIF) neurons with common fluctuating input where each neuron's membrane voltage, $V_i (1 \leq i \leq N)$ evolves as follows:

$$\tau_m V_i' = -V_i + \Delta_T e^{\frac{V_i - V_S}{\Delta_T}} + I_i(t), \quad I_i(t) = \sqrt{\sigma^2 \tau_m} \big[\sqrt{1-\lambda}\xi_i(t) + \sqrt{\lambda}\xi_c(t)\big]. \tag{2}$$

Notably, it has been shown that pairwise maximum entropy (PME) models are incapable of estimating this network very well especially when the firing rate of neurons is low, neural correlations are high, or the population size is large Amari et al. (2003); Macke et al. (2011b); Leen & Shea-Brown (2015). Moreover, experimental studies show the existence of higher order correlations in the brain Binder & Powers (2001); Khuc-Trong & Rieke (2008). Therefore, the network above has been used as a tool to assess the performance of methods for capturing beyond pairwise correlations Macke et al. (2011b); Leen & Shea-Brown (2015). Similarly, we generate multiple data sets using varying parameters to get different values for the mean activity $\mu$ and (pairwise) correlation $\rho$ and compare the goodness of fit between $nSPN^{N/2}$ and PME models for population size of $N$ (all parameter values in Supplementary Material). Note that $nSPN^{N/2}$ is chosen to have the same number number of parameters as the PME, $N(N+1)/2$ ($nSPN^{N/2}$ has $N^2/2 - 1$ free parameters to be more precise).

Pairwise Maximum Entropy models have minimal assumptions about the spiking pattern of the population beyond the firing rate of each neuron and the relationship between each two neurons. In full generality, a PME model is complex with many parameters, and is computationally expensive in general. In fact, as mentioned before, an exact fit of PME is NP-hard. Therefore, we instead fit to the probability that the a population of $N$ neurons will generate $k$ spikes overall in a single time bin, i.e.:

$$P_{\text{PME}}(k) \propto \binom{N}{k} \exp\big(\alpha k + \beta k^2\big) \tag{3}$$

where $\alpha$ and $\beta$ are free parameters, obtained from fitting the model to population activity. Importantly, instead of fitting the PME to the whole joint distribution we obtained $\alpha$ and $\beta$ by directly optimizing the above equation. This measure reflects the maximum possible power of any PME model without any need to actually obtain all variables (it essentially provides a bound on the performance of the full model). Note that, the distribution of average population activity fully reflects the goodness of fit in fully homogeneous populations Roudi et al. (2009a). However, there is no guarantee that existing approximation methods for PMEs achieve such fit.

As opposed to the PME, we fitted an $nSPN$ to the whole joint distribution (200 iterations of Expectation Maximization (EM) implemented in SPFlow library Molina et al. (2019)), giving the PME a huge advantage. Specifically, the difference between the model's prediction and the EIF (total) population spike distribution is quantified with the Jenson-Shannon (JS)-divergence of $\frac{1}{2}D\big(P_{model}(k)\|M(k)\big) + \frac{1}{2}D\big(P_{EIF}(k)\|M(k)\big)$ where $M$ is the averaged distribution, i.e. $M(k) = \frac{1}{2}P_{model}(k) + \frac{1}{2}P_{EIF}(k)$ and $D(.\|.)$ is the Kullback–Leibler (KL)-divergence. Since JS-div scales with $\log(N)$, we report the normalized value, i.e. JS-div$/log(N)$. We set $\lambda = .30$ and .59 to make

the pairwise-correlation $\rho$ equal to .1 and .25 respectively, keeping other parameters the same as above. In both cases, the mean activity rate $\mu$ is .1. The number of neurons in the network is $N \in [8, 16, 32, 64, 128]$, and each data point is obtained from training on 100000 and testing on another 100000 samples. As shown in the left panel of figure 2, nSPN models the EIF network significantly better than PME (the JS-divergence is orders of magnitude lower) especially when the number of neurons and correlation ($\lambda$) are high (dotted lines $\lambda = .59$, solid lines $\lambda = .3$).

We also compared our method to other approaches for capturing joint probability distributions of spiking activity. We trained a Bernoulli Restricted Boltzmann Machine (RBM) with $N$ visible and $N/2$ hidden units ($N^2/2$ free parameters). The JS-div of the RBM was an order of magnitude higher than the SPN (figure 2, left plot). Moreover, the SPN scales better with the increase of $N$ (Note that the y-axis is in log-scale). Importantly, the RBM is one of the best known latent-variable model in capturing population correlations (Köster et al., 2014).

One of the goals of these benchmarks is to assess model performance as a function of population size ($N$). Hence in some simulations we include a single, global input to all neurons (governed by parameter $\lambda$ in the equation below). In addition, we introduce correlated subpopulations by adding a common input to specific subpopulation both pairwise (via $\lambda_2$) and in quadruples (via $\lambda_4$).

$$I_i(t) = \gamma + \sqrt{\sigma^2 \tau_m} \big[ \sqrt{1 - \lambda} \xi_i(t) + \sqrt{\lambda - (\lambda_2 + \lambda_4)} \xi_c(t) + \sqrt{\lambda_2} \xi_{i,2}(t) + \sqrt{\lambda_4} \xi_{i,4}(t) \big]. \quad (4)$$

For simplicity, we assumed these common inputs are among adjacent nodes: $\forall 1 \leq k \leq N/2 :$ $\xi_{2k-1,2}(t) = \xi_{2k,2}(t)$, and $\forall 1 \leq k \leq N/4 : \xi_{4k-3,2}(t) = \cdots = \xi_{4k,2}(t)$. We repeated the previous set of fits ($\lambda = .59, .3$), this time with $\lambda_2 = \lambda_4 = .05$. As demonstrated in figure 2 middle left plot, SPN outperformed the other models by orders of magnitude again.

We also performed simulations with heterogeneous inputs, resulting in heterogeneous population activity. Neurons are divided into four groups with different voltage reset values and different values of $\lambda$ (values can be found in the supplementary material). This network was simluated with both $\lambda_2 = \lambda_4 = 0$ (Figure 2 middle-right: dotted lines) and $\lambda_2 = \lambda_4 = .05$ (Figure 2 middle-right: solid lines). Our SPN significantly outperformed the other models. Note that due to the heterogeneous network activity, the PME direct fitting results are not a valid lower-bound for JS-div of a pairwise model anymore.

We also tested recurrent networks with heterogeneous connectivity. We analyzed a well-known connected "balanced" network of excitatory and inhibitory spiking neurons Brunel (2000). The dynamics of the membrane voltage of each neuron in this network is:

$$\tau_m V_i' = -V_i + RI_i(t), \quad RI_i(t) = \tau_m \sum_j J_{ij} \sum_k \delta(t - t_j^k - D), \quad (5)$$

where $I_i(t)$ are the synaptic currents caused by spikes arriving at synapses of neuron $i$ (more details in the supplementary material). We specifically fit the models to two stable setups (figure 8, c and d) of the original paper with "stationary" and "slow oscillation" global activity Brunel (2000). As shown in the right plot of figure 2, JS-div of the SPN was orders of magnitude smaller than other methods. Importantly, here the size of the networks were always 1000, and each fitting was on the first $N$ excitatory neurons. Moreover, similar to our previous network, JS-div from fitting PME directly to the population activity is no longer a valid lower-bound for pairwise models.

EXPERIMENTAL NEURAL RECORDINGS

To assess our method on experimental neural recordings we modelled the neural data from the Neuropixels Visual Coding data set of the Allen Brain Observatory (https://observatory.brain-map.org) (Siegle et al., 2021). In this data set, different stimuli (Gabors, flashes, drifting gratings, etc) were shown to 26 mice while the neural activity of their visual cortical and sub-cortical regions were recorded simultaneously with multiple Neuropixels probes. We modelled the neural population structure during the viewing of drifting gratings in 4 directions ($[0°, 45°, 90°, 135°]$) with the contrast level of .9. Each of these 4 stimuli was shown to 26 mice 75 times. We analyzed the neural activity (spikes) of six regions in the visual cortex: VISp, VISl, VISal, VISpm, VISrl, VISam. Following other studies analyzing neural correlations Köster et al. (2014); Tkačik et al. (2014), the time bin was set to $\Delta t = 20$ms, balancing for minimizing spike loss due to binning, and accounting for activity delays between neurons.

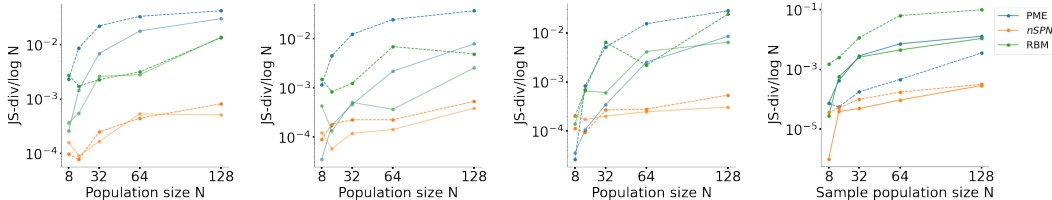

Figure 2: **Simulation of neural population activity confirms the practicality of our SPN-based framework to measure correlations among neurons.** Left: Normalized JS-divergence versus population size for two conditions with similar mean activity rate ($\mu = .10$) but different correlations (dotted lines: $\lambda = .59$, solid lines: $\lambda = .30$). $nSPN$ outperformed both RBM, as well as the best possible PME. Middle left: Similar to the left plot, but with additional common inputs to pairs and quadruples ($\lambda_2 = \lambda_4 = 0.05$; dotted lines: $\lambda = .59$, solid lines: $\lambda = .30$). Middle right: Same comparison on a heterogeneous network with or without local inputs (dotted lines: $\lambda_2 = \lambda_4 = 0$, solid lines: $\lambda_2 = \lambda_4 = 0.05$). Right: Same analysis on $N$ excitatory neurons of a well-known sparsely connected network with total population size of 1000 in two regimes, stationary global activity with irregularly firing neurons (dotted lines), and slow oscillation of global activity with very rare irregular firing (solid lines).

Each stimulus duration was 2 seconds. We analyze the response from $400ms$ after stimulus onset to $800ms$ after stimulus onset. The delay is to make sure the corresponding signal has reached the neurons in all areas. Moreover, individual neurons' change upon exposure to the same sensory stimuli (via adaptation), hence the short period. For each grating direction, we concatenated data from 5 presentations of $400ms$ to get 100 data points. The order of presentation of grating directions is random. In order to minimize the effects of state changes in the animal, we concatenate trials from subsequent presentations of the same grating direction. This data processing results in 15 blocks of population activity per each tuple (mouse, direction, area), each of which we modelled with a $nSPN^l$. Notably, we used the same set of hyper parameters for all fits. Hyper parameter tuning of $nSPN^l$ for each fit via cross validation improved the overall fit only slightly, demonstrating the robustness of our method.

We also fit our simplest structure $nSPN^1$ (Naive Bayes) in order to have a correlation-free baseline and compute the difference, $\Delta ll_{SPN}$. We fit both models on a modified version of data in which we shuffle the spikes of each neuron in the $400ms$ of single stimulus showing for the primary visual cortical population (as the most sensitive area to events). We observed a large difference between $\Delta ll_{SPN}$ in the original and shuffled data. This difference confirmed our assumption about $\Delta_{SPN}$ in the original data is mostly the product of neural correlations (Figure 3 left).

Since the population is not homogeneous, this time we used Minimum Flow Probability (MFP) (Sohl-Dickstein et al., 2011) that approximates pairwise maximum entropy variables through a direct fit to the data (implementation of (Lee & Daniels, 2019)). To the best of our knowledge this is the most accurate methods of pairwise approximation. We also fitted an RBM to the data. The number of hidden units in the RBM was determined through 10-fold cross-validation within the 100 samples.

The (total) population activity is not an ideal measure for performing model comparison. However, given the number of samples, i.e. 100, more desirable measures are prohibitive. Compared to other models, $nSPN$ produced significantly lower normalized JS-div for all of the 6 regions (Figure 3, middle plot). In fact, J-div of $nSPN$ was lower than other methods for nearly every single (animal, area, direction). The right plot in figure 3 shows this comparison v.s. RBM.

We also tested our model on four short (340 ms) movie clips, cut from the "natural movie one" stimuli of the same data set, which is a one-shot 30 second video repeated 60 times. The frame rate is 30 Hz, making each of our clip consists of 10 frames, starting from frame number 240, 390, 490 and 890. Similar to the previous stimulus, we grouped 5 consequent clips together, which resulted in 12 blocks of population activity. The time bin was set to 17 ms, meaning that each block included 100 data points. We repeated the analysis described above for this stimulus type and got consistent results (Figure 4). Notably, $\Delta ll_{SPN}$ was higher in general in this stimulus, probably because of longer time difference (at least 30 seconds) among the 5 trials of each group. Because of this, very high computational cost, and poor results on a simpler stimulus type, we did not include $MPF$ in

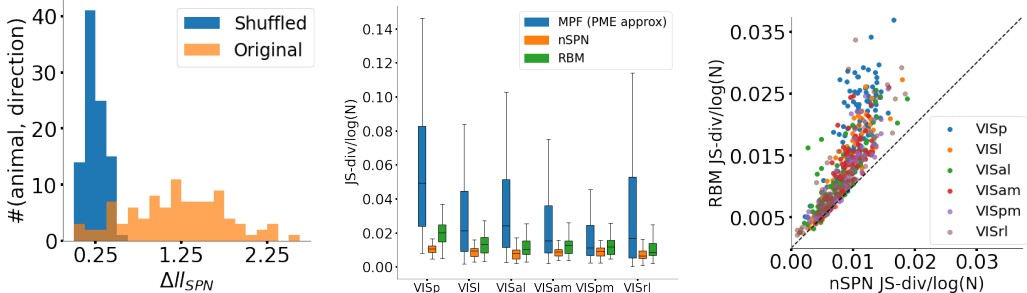

Figure 3: **Modeling high order correlations in neural recordings during drifting grating stimulus.** Tested on large-scale Neuropixel recordings from mouse visual cortex upon exposure to during drifting grating stimulus, $nSPN$s explains neural population structure and correlations in different regions of mouse brain significantly better than pairwise entropy model estimation (MPF) and RBM. Left: Distributions of $\Delta ll_{SPN}$ by experiment for the unshuffled (orange) vs shuffled (blue) spike distributions. Middle: Box plots of the distribution of normalized JS-divergence over experimental sessions by visual cortical area. Right: Comparison of JS-divergence of SPN with RBM for every single (animal, area, direction) (each dot).

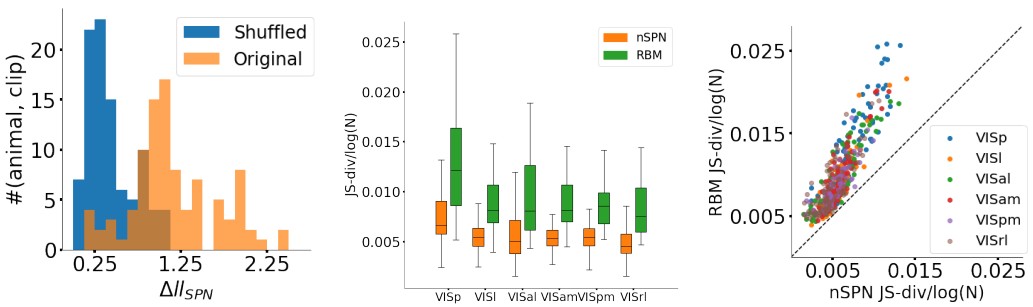

Figure 4: **Modeling high order correlations in neural recordings during natural movies.** Same analysis discussed in figure 3 but on four different natural movie clips. Left: Distributions of $\Delta ll_{SPN}$ by experiment for the unshuffled (orange) vs shuffled (blue) spike distributions. Middle: Box plots of the distribution of normalized JS-div over experimental sessions by visual cortical area. Right: Comparison of JS-div of SPN with RBM for every single (animal, area, movie clip) (each dot).

our analysis for this stimulus. Moreover, as observed in the right plot of figure 4, superiority of our model is bolder compared to the simpler stimulus of drifting gratings (right plot of figure 3).

FURTHER ANALYSIS ON THE ROLE OF NEURAL CORRELATIONS

Without knowledge of the ground truth, data interpretation is much harder especially when the data is limited and high dimensional. To further investigate the role of (spatial) neural correlations in population activity we extended our analysis to dynamic latent models that look at each trial as a whole (Macke et al., 2011a; Pfau et al., 2013; Gao et al., 2015). Therefore, these models take long temporal correlations into account. While these models are usually applied to motor tasks with longer trial duration, temporal correlations could affect the population structure in our tasks as well, especially during the natural movie stimulus. We fit a Poisson Linear Dynamical System (PLDS) with nuclear norm penalized rate estimation (Pfau et al., 2013) [1] to the population activity of primary visual cortex (VISp) as example extensions of our analysis during both natural movie clips and drifting gratings similar to our previous fits. The number of latent factors were determined by leave-one-out-cross validation over trials (which were 5). As shown in the left plot of figure 5, JS-divergence of $nSPN$ were lower for every single stimulus and mouse except one (animal, stimulus) pair, especially in the more complex stimuli, i.e. movie clips. Notably, latent dynamical systems have demonstrated the ability to capture pairwise correlations, in addition to temporal dynamics of the

---

[1]code from: `https://bitbucket.org/mackelab/pop_spike_dyn`

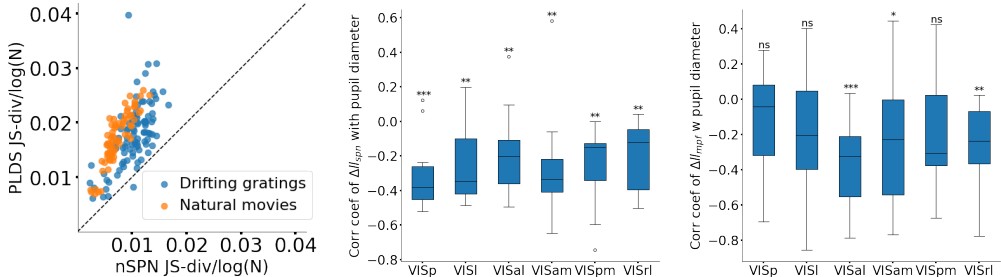

Figure 5: **Further analysis on neural population correlation.** (left) Poisson Linear Dynamical systems which model temporal evolution and correlations of neural activity explain VISp population structure worse than $nSPN$ in almost all mice and stimulus conditions. (middle) The Pearson correlation between $\Delta ll_{spn}$ and pupil diameter over experiments is significantly negative in visual cortical areas. (right) Same approach applied on MPF, $\Delta ll_{mpf}$ can not fully capture this phenomenon.

system (e.g. temporal cross correlations in (Macke et al., 2011a). Therefore, this result suggests that higher order short-term/spatial correlations (20 ms) plays a more crucial role here. Using mixtures, $nSPN$ can capture (discretized) temporal dynamics but (probably) not as good as methods with explicit components of modelling dynamics.

Our second attempt on checking the validity of our conclusions involved previous neuroscience literature. Many experiments have suggested that task engagement in the form of attention and arousal has the effect of reducing neural correlations, decreasing the level of synchrony (Gandal et al., 2012; Uhlhaas et al., 2009), and even measurable behavioral changes (Cohen & Maunsell, 2009). These experiments are mostly based on very noisy data such as the Local Field Potential (LFP) and rough estimates of signal frequency (Gandal et al., 2012; Pfeffer et al., 2022; Vinck et al., 2015). Having a quantitatively reliable measure based on spiking data, i.e. $\Delta ll_{SPN}$, we examined the effect of arousal during drifting grating viewing. Specifically, in each animal we looked at the (ratio to minimum) pupil diameter (a common measure of arousal in mice) changes at each of the chunks described above for all directions. To alleviate the effect of trail-to-trial variability (e.g. caused by behavioral state change) on neural correlations we removed (mouse, direction) pairs with $\Delta ll_{SPN}$ of more than .25 in the primary visual area for the shuffled data. The mean distribution of Pearson correlation coefficient between pupil diameter and $\Delta ll_{SPN}$ across all animals was significantly below zero for all regions as shown in Fig. 5 (left). To the best of our knowledge, even with LFP data, there is no work demonstrating the effect of arousal on higher visual areas during an experiment (See (Vinck et al., 2015) for VISp).

We repeated the same process for MPF by using the difference between log-likelihood of the full (first and second order) model with the first-order only version (second order variables set to zero). The result was significantly different (Fig. 5, right). Importantly and as mentioned, previous literature results (especially on VISp) are strongly in favour of the SPN results. Importantly, the stimulus in the previous VISp study was also drifting gratings (Vinck et al., 2015). Moreover, this result further demonstrates the inability of the PME in modelling the full joint population activity (as opposed to the total population activity).

## 5 DISCUSSION

Correlations are an important channel of information in neural activity, with strong consequences for coding properties. Using SPNs, we have constructed a computationally efficient approach to modeling structure and correlations in populations of spiking neurons. Tested on simulated and experimental data, our approach outperformed both energy-based and latent variable methods. Due to the nature of our scientific question in this paper, i.e. population correlations, we focused on short time spans in which the external variables and animal state remain constant. Our framework, however, has the capacity to be applied more broadly in computational neuroscience, for example to capture population dynamics in different time scales or as a de-mixing tool for coded features in individual or population of neurons.

**Reproducibility**: The data set used in this paper, Allen Brain Observatory, is publicly available online. All parameters of the simulated data has been specified in the result section and/or the supplementary material. Similarly, all parameters related to the methods are mentioned in the results section and/or the supplementary material. In addition, all the code is based on open-source repositories, cited in the results section. Finally, the code for data is available in `https://github.com/koosha66/NeuralSPN`.

**Acknowledgment**: We wish to thank the Allen Institute founder, Paul G. Allen, for his vision, encouragement and support.

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
