# OpenReview forum: "Efficient approximation of neural population structure and correlations with probabilistic circuits"
_ICLR.cc/2023/Conference — ICLR 2023 poster_

### Official Review · Reviewer_gQaa · 2022-10-24

**Confidence:** 4
**Correctness:** 4
**Technical Novelty And Significance:** 4
**Empirical Novelty And Significance:** 4
**Recommendation:** 8

**Clarity, Quality, Novelty And Reproducibility:**

The paper is well written, and the work is of high quality and
novel. The method seems explained at a sufficient level of detail.

**Strength And Weaknesses:**

## Strengths
1. The proposed method is novel and highly original.
2. The comparisons with synthetic and recorded data convincingly show
   that at least in the examples analyzed here the method outperforms
   some notable baselines.
3. The paper also does a good job of introducing sum-product networks
   to a potentially broad audience.

## Weaknesses
In my view this is a generally solid paper; however, here are some
suggestions for strengthening the paper further by increasing coverage
of related literature.
1. When discussing comparisons to existing approaches to model neural
   correlations, it could be appropriate to mention recent works on the
   statistical complexity of maximum entropy models for binary data
   (Beretta et al 2018, De Mulatier et al 2020). These have shown that
   the complexity of a model depends not so much on the order of the
   interaction but more on their mutual arrangement, and also that it
   is possible to efficiently search through the space of "simple"
   maximum entropy models. I wonder if anything is known about
   potential efficient approximations of the Bayesian evidence of
   sum-product networks that could be used in a formal model-selection
   context.
2. When discussing neural correlations from a neuroscience perspective
   and its relationship with brain state, consider citing Cohen and
   Maunsell 2009, which has shown that attention modulates pairwise
   correlations in monkey visual cortex, and that this modulation of
   population structure has substantial behavioral implications.
3. Another class of approaches aimed at uncovering the statistical
   structure of population data that seem worth mentioning is that
   based on GLMs (Pillow et al 2008). Despite their limitations, these
   methods scale well to large populations and are applicable to
   complex experiments involving rapidly-varying external variables
   (see e.g. Runyan et al 2017).
4. Moreover, it would be great if the authors could add the K-pairwise
model (Tkacik et al 2014) to their comparisons in Figure 3 and 4. That
model, which is already cited in the present paper, is considered a
particularly powerful (while practical) higher-order extension of
pairwise maxent models and seems like a good candidate to benchmark
against. However this is just a suggestion that I think could make the
paper's message even stronger, not something necessary for acceptance.

### Typos:
- page 5, bottom: "if and only OF all neurons are independent..."
  (of→if)
- page 7: "The RBM performs better with SMILE size populations."
  (smile→small)
- page 7: "their average log-likelihood difference (...) increases
  with γ". I think here γ should be replaced by λ.
- please check for consistency of the symbol used for "milliseconds"
  (or seconds). Sometimes it's ms, sometimes msc, sometimes msec. The
  correct SI abbreviation is ms (s for seconds).

### References:

Beretta, Alberto, Claudia Battistin, Clélia De Mulatier, Iacopo Mastromatteo, and Matteo Marsili. 2018. “The Stochastic Complexity of Spin Models: Are Pairwise Models Really Simple?” Entropy 20 (10): 739. https://doi.org/10.3390/e20100739.

Mulatier, Clélia de, Paolo P. Mazza, and Matteo Marsili. 2021. “Statistical Inference of Minimally Complex Models.” arXiv. https://doi.org/10.48550/arXiv.2008.00520.

Cohen, Marlene R., and John H. R. Maunsell. 2009. “Attention Improves
Performance Primarily by Reducing Interneuronal Correlations.” Nature
Neuroscience 12 (12): 1594–1600. https://doi.org/10.1038/nn.2439.

Pillow, Jonathan W., Jonathon Shlens, Liam Paninski, Alexander Sher, Alan M. Litke, E. J. Chichilnisky, and Eero P. Simoncelli. 2008. “Spatio-Temporal Correlations and Visual Signalling in a Complete Neuronal Population.” Nature 454 (7207): 995–99. https://doi.org/10.1038/nature07140.

Runyan, Caroline A., Eugenio Piasini, Stefano Panzeri, and Christopher D. Harvey. 2017. “Distinct Timescales of Population Coding across Cortex.” Nature 548 (7665): 92–96. https://doi.org/10.1038/nature23020.

Tkačik, Gašper, Olivier Marre, Dario Amodei, Elad Schneidman, William Bialek, and Michael J. Berry. 2014. “Searching for Collective Behavior in a Large Network of Sensory Neurons.” PLOS Computational Biology 10 (1): e1003408+. https://doi.org/10.1371/journal.pcbi.1003408.

**Summary Of The Paper:**

This paper presents a novel method for modeling neural correlations
based on sum-product networks. The idea is that inference of the
statistical structure of neural population activity is made easier by
the assumptions encoded in the choice of this type of network, which
enforce a hierarchical structure of alternating mixtures and
factorizations. Within this class of models, the model structure is
chosen with a procedure based on hierarchical clustering computed from
an information metric. Once the model structure is fixed, model
parameters are learned by EM. The paper introduces sum-product
networks, describes the proposed method, and compares it with existing
methods on synthetic data as well as on neural recordings.

**Summary Of The Review:**

This is a good paper. The proposed technique is novel and original,
and is shown to be effective at capturing the structure of neural
population activity. The theory is well explained and the concrete
benchmarks are convincing.

---

> ### Author Response · Authors · 2022-11-19
> **Thank you for your feedback!**
>
> We thank the referee for their strong review of our work and their comments. We appreciate the pointers to more recent literature on maximum entropy models as well as other relevant literature.  Our final revision will make connections with these papers.  Time permitting, we will also add the Tkacik et al 2014 model to our comparison. Notably, even if the Tkachik et al 2014 model works well, it is computationally very expensive due to its reliance on the pairwise-Ising model.

---

### Official Review · Reviewer_Ae2s · 2022-10-24

**Confidence:** 3
**Correctness:** 3
**Technical Novelty And Significance:** 3
**Empirical Novelty And Significance:** 3
**Recommendation:** 8

**Clarity, Quality, Novelty And Reproducibility:**

**Novelty**: As stated above, to my knowledge, the paper explores a novel application of SPNs. SPNs have been developed in prior works, a technical novelty of this paper is the method for learning correlations in neural populations.

**Reproducibility:** Training details for baseline methods are currently missing, e.g., it is unclear how RBMs were trained exactly. The authors write that the code will be made available upon publication; unfortunately it was not possible to look these details up.

**Clarity**: For the most part, the paper is clearly written.

**Minor comments:** Adding citations in brackets would improve readability. "smile size" -> "small size".

**Strength And Weaknesses:**

**Strengths:** The use of SPNs for neural population analysis of spike train data is, to the best of my knowledge, a novel application. On problems and metrics considered in the paper, results overall show promising improvements over the baselines considered.

**Weaknesses:** To me, the biggest weakness of the paper is its narrow empirical evaluation -- the authors only consider one example in which ground-truth is known. I would have hoped for more extensive evaluation on problems with synthetic data, in particular, considering heterogenous networks with known ground truth. In addition, I found a discussion of SPNs limitations for neural data analysis to be missing.

**Summary Of The Paper:**

The paper proposes the use of Sum-Product Networks (SPNs) to study spike train recordings of populations of neurons. It compares SPNs against latent variable and energy-based methods on a synthetic problem: Correlations are estimated for a network of homogenous exponential integrate-and-fire neurons. It also applies SPNs to experimentally obtained neural recordings from the Allen Brain Observatory.

**Summary Of The Review:**

The application of SPNs to neural population analysis is interesting and I am surprised this has not been explored previously. The results are promising, but empirical evaluation on synthetic examples with known ground truth is currently limited to a just a single problem. I think that a more thorough empirical analysis and a discussion of limitations would be critical for practioners considering application of SPNs to their data analysis problems.

### Update after rebuttal

I appreciate the addition of new experiments and updated my overall recommendation.

---

> ### Author Response · Authors · 2022-11-19
> **Thank you for your feedback!**
>
> We appreciate your review and comments. We have expanded the comparisons with simulations in a few ways. First, we have included networks with heterogeneous inputs to produce both overall heterogeneous activity patterns and heterogeneity across the different pairs and quadruples that receive correlated input.  Second, we have also included simulations of networks with heterogeneous connectivity.  For this, we have used the E-I balanced networks from Brunel (2011).  Our approach shows similar performance with these simulations as well.
>
> We have also expanded the application of the approach to real neural data, specifically including responses to natural movies. Our results indicate that the model performs well in this case.
>
> We are planning on adding a discussion on the limitations of this method for neural data analysis to the paper. During the rebuttal period, we have focused on adding more methods, simulations, and experimental data analysis.

---

### Official Review · Reviewer_Lrts · 2022-10-25

**Confidence:** 4
**Correctness:** 2
**Technical Novelty And Significance:** 3
**Empirical Novelty And Significance:** 2
**Recommendation:** 6

**Clarity, Quality, Novelty And Reproducibility:**

The paper is well-written. The quality of the work is marginal. The application of SPN models to fitting correlated neuronal data is novel. The data set used in the current study is open source. It is not clear to me whether the model of the current work would be released.

**Strength And Weaknesses:**

Strength:
Compare to existing methods such as pairwise maximum entropy models, the SPN model is more computationally efficient and fits neuron spiking better.
Weakness:
1.	The interpretability of the current model is low.
The authors show that the SPN models could use correlation information in predicting population neuron activities, as the current model performs better than the naïve Bayesian model which does not model neuronal interactions. However, it is not clear to me how to interpret higher-order correlations using the current model. Which part of the model learns the higher-order correlation? Can we read out pairwise correlation from the model parameters?
2.	Not well-supported interpretation
The authors suggest that the delta_ll value is a measure of ‘correlation’ (Fig. 5), which I found not well supported. First of all, the relationship between delta_ll and higher-order/linear correlations is not well established. In particular at the physiological neuron population. Second, it is a very low-resolution measure as one value (delta_ll) is computed for one dataset while the biological physiology (eg. pupil dynamics) changes much faster. Third, the fact that a control model (MPF) does achieve similar results on the majority dataset makes me confused even more about this measurement.
3.	Model is only tested with simple data
The simulation data seems to only have a simple correlation structure (eq. 3). The correlation of the simulated neuron population is generated through common input, with no synaptic interactions or coupling. Also, it seems the whole population is receiving the same common time-varying input, which would probably make the correlation structure super simple. Plus the correlation between the simulation set is pretty high (>0.1). In addition, the real data set is also a simple dataset: visual neuron response evoked by four drifting gratings. Such a dataset is expected to have a low-dimensional correlation structure. It is not clear to me whether the model performance on these simple datasets would translate easily to more broad applications. Plus, other existing approaches like GLM might work as well on these simple data.


**Summary Of The Paper:**

Learning the correlation structure of neuron population is tuff. Usually, only pair-wise linear correlations can be measured. It is hard to compute higher-order correlations in the population. The authors design a Sum Product Network (SPN) to learn higher-order correlations of a neuron population. Overall, the paper is well written, however, I feel not convinced that the current model could have a larger application.

**Summary Of The Review:**

I feel excited reading the first half of the paper, as the authors set up the problem and direct the audience very well. However, the result section is a bit disappointing. The keyword ‘higher order correlation’ is repeatedly mentioned in the paper, however, only indirect measurement (delta_ll) is provided to suggest that the model learns the correlation. Without more information about the model and better validation of the model, it is really hard to generalize what has been presented in the paper.

---

> ### Author Response · Authors · 2022-11-19
> **Thank you for your feedback!**
>
> We thank you for your comments and appreciate that the first half of our paper excited you.  We hope to extend that excitement to the remainder of our paper and aim for the following points below to push in that direction.
>
> We appreciate your comment about interpretability. Our model, similar to the Ising model, seeks to model the distribution of binary words produced by a neural population. You are correct that the SPN does this in a way that makes it difficult to immediately extract specific orders of correlations. It is not, for example, like a maximum entropy approach that attempts to fix specific moments of the joint spike distribution.
> Instead, it attempts to capture the highest correlations/synchronous activity of sub-populations. In fact, this is the main reason that it can model the population structure so well. SPN is more similar to the sparse Ising model (Ganmore 2011) which is based on frequent patterns. Unlike the sparse model, however, SPN does not rely on high sparsity of the data and many samples.
>
> To compute any specific n-point correlations, you would need to marginalize the graph, which can be done in linear time (this is the main advantage of the SPN). To compute the marginalized joint activity, both $I(s_i)$ and $I(\bar{s_i})$ is set to 1 for marginalized neurons, $n_i$. For example, the joint probability distribution of $p(s_1, s_2)$ in figure 1 is calculated by setting both $I(s_3)$ and $I(\bar{s_3})$ to 1 in all four configurations of $s_1, s_2$ (00, 01, 10, 11) and computing the root value in each of these configurations.
>
> In a shallow graph, specifically, the graph representing a Naive Bayes model there are no correlations.  One way of describing this in terms of the graph is that every neuron appears within the scope of distinct "sum" nodes.  In deeper graphs, separate neurons may appear within the scope of the same intermediate "sum" nodes, which is an indication that there will be a non-trivial contribution to the marginal joint probability of those neurons, i.e. they will be correlated.  The exact numerical quantity is not easy to discern from the graph, but the structure will provide an indication of where the joint probabilities of $n$ neurons will deviate from independence.
>
> We regard $\Delta_{ll}$ as a measure of correlation because it is the relative log-likelihood of the fit graph with a Naive Bayes, i.e. independent, model.  By construction, were the neurons all independent, then $\Delta_{ll}$ would be zero.
> This is similar to the common approach in pairwise Ising models where the KL-divergence ratio is calculated (e.g. see Roudi and Abbot 2011). Kl-divergence of the whole distribution, however, is not easy to compute when the data is high-dimensional. Notably, we can use the log-likelihood of the whole population activity only because our model calculates real (normalized) probability. This is not possible with other methods.
>
> We agree with the referee that the paper would be improved with a broader range of simulations.  Regarding the use of common input, the important factor for our purposes is the generation of correlations in the activity, regardless of whether from common input or via interactions.  We have included now both simulations with heterogeneous common input (as well as heterogeneous pairwise and quadruple-wise correlated input) as well as a simulation of a heterogeneously connected network, namely an E-I balanced network from Brunel (2000).
>
> For the actual neural data, we have included analyses on not only grating responses but also natural movie responses.
> Our results are actually better in the movie responses (compared to RBM and PDLS) because it is a more complex stimulus. Regarding the performance of MPF, we do believe that this is actually reasonable as pair-wise correlations are by definition more prominent in the data. We are showing that they are not telling the whole story though. Also, note that we included all neurons that were even slightly active during any stimulus condition as arbitrary data analysis choices may change the outcome drastically. This means that in each of the four conditions (direction), there are many inactive neurons. This is extremely in favor of weaker methods as all methods perform very well in "predicting" all 0s.

---

> > ### Comment · Reviewer_Lrts · 2022-11-27
> > **Thanks for your response**
> >
> > Thanks for detailed response to my comments! I agree increase the recommendation of the current paper. Regardless its weakness, I still see the value of sharing the work with broader audiences. Plus, other reviewers seem to appreciate the work.

---

### Official Review · Reviewer_S4f9 · 2022-10-25

**Confidence:** 3
**Correctness:** 3
**Technical Novelty And Significance:** 3
**Empirical Novelty And Significance:** 3
**Recommendation:** 8

**Clarity, Quality, Novelty And Reproducibility:**

### Clarity
Despite being very familiar with the topic of modeling neuronal correlations, I had a fairly hard time following the paper, mostly because I'm not very familiar with SPNs and the paper didn't guide me through the material in an intuitive way.

### Quality
I have no doubts that the modeling is technically solid, but I find it hard to conclude whether the approach is indeed as strong as claimed by the authors because of the missing comparisons with alternative strong approaches.

### Novelty
Judging novelty is somewhat difficult, mostly because I'm not familiar with the SPN framework, and neither the limitations of existing work in that direction nor how the authors overcome them are clearly discussed in the paper.

### Reproducibility
On the positive side, the paper contains an explicit reproducibility statement and promises to published the code.


**Strength And Weaknesses:**

### Strengths

 + Estimating correlations between neurons from limited data is a long-standing, but not completely solved problem
 + Approach seems promising and novel (to my knowledge), at least within the application domain
 + Explicit reproducibility statement and promise of code being published


### Weaknesses

 1. Important baselines for modeling neuronal correlations missing
 1. Relationship to prior work on SPNs somewhat unclear


### 1. Baselines

Although the two baselines chosen by the authors (PME & RBM) are sensible choices, several alternative methods are missing. For instance, the literature on noise correlations has shown that common mode or low-rank fluctuations are abundant. To model such effects, latent variable models have been developed and yield state of the art performance. It is not clear to me why the authors consider an RBM "the best latent variable model" (p.2) - at least Koster et al. (2014) don't show that. Examples for alternative latent variable models include Poisson Linear Dynamical Systems (Macke et al. NeurIPS 2014) or deep gamma dynamical systems (Guo et al. NeurIPS 2018). Also relevant: Pfau et al., NeurIPS 2013, Ramesh et al., NeurIPS 2019 Neuro-AI workshop or very simple baselines like exponential family PCA or Factor Analysis. As it stands, it is not possible to tell from the evidence presented whether or not the authors' model is competitive.


### 2. Relationship to prior work on SPNs

On p.3 the authors comment that "there exist few algorithms for structure learning Gens & Pedro (2013); Vergari et al. (2015)," but they developed their own method. Why not use existing methods? How does the method developed here differ from existing methods and how does it compare in terms of performance? While I am not familiar with the literature on SPNs, the present paper seems to mostly do knowledge transfer from other fields and apply an existing method in a new context. It doesn't become clear, however, where the authors make novel contributions because existing approaches from other fields are not applicable and need to be modified, and what these modifications are.


### Other

- On p.7 the authors comment that their "model's main target is to capture the largest correlations with its available resources (free parameters)." Why is this a reasonable thing to do in a model where all pairs are expected to have the same correlations due to the common input?

- Related to the previous point, it is not clear to me why in a model where all correlations are driven by a single common input, the authors model should outperform an RBM by two orders of magnitude. What's the mechanism of this huge improvement? In such a simple data-generating process, I would expect any model with a single common mode to perform quite well. What additional inductive bias does the authors' model have that other baselines don't have?

- I could not follow the description of how the authors partition the data on the bottom of p.7 / top of p.8. What does it mean to "combine[d] the data of 5 consequent presentations with the same direction"? How does this "keep the state of the animal [...] as unchanged as possible"? Why does this processing result in 15 blocks? Blocks of what? Relatedly, I couldn't follow the bottom of p.8 and what exactly Fig. 4, left, shows (e.g. what's the difference between "SPN" and "SPN CV"?)


**Summary Of The Paper:**

The authors propose a method to model correlations between neurons in large populations which is based on sum-product networks (SPNs). They show that it outperforms two baselines (pairwise maximum entropy models and restricted Boltzmann machines) on a synthetic dataset and the Allen Brain Observatory neuropixels dataset.


**Summary Of The Review:**

Potentially interesting and novel (within application domain) approach, whose performance is somewhat difficult to judge due to limited comparisons with existing work.

### [Update after rebuttal]

The authors have included an additional baseline and clarified their contributions w.r.t. SPNs. I now support the paper's publication.

---

> ### Author Response · Authors · 2022-11-19
> **Thank you for your feedback!**
>
> We appreciate your review and comments. We particularly appreciate the links to the literature on latent variable modeling.
>
> The comment about baselines is an important one.  We have added an analysis using the Poisson Linear Dynamical System model from Pfauer NeurIPS 2013. We have applied this model to our analysis of real data. (Due to time constraints, we have only applied this to real data, VISp area, 3 stimulus/condition pair) for now; we will extend this to a comparison with other data, region, and stimulus/condition pairs as well.)  The new analysis has been added to Figure 5 (left panel). One can see that our SPN model outperforms the PLDS model in every mouse and stimulus/condition pair.
>
> The referee is also correct to ask questions about structure learning methods for SPNs.We initially developed our method due to the simplicity and interpretability of the process, as well as its natural connection with shallow networks. Moreover, there is more control over the free parameters in our initially proposed approach. Having been developed for large machine learning data sets, these methods often ignore many dependencies by forcing very loose tests/thresholds. Finally, common clustering methods integrated into these approaches, e.g k-means in learnSPN, are often not suitable for binary highly-sparse data. Upon further investigation we discovered that at least for our current data sets learnSPN works well and actually slightly outperforms our method (albeit slightly slower) if we choose a stricter than usual independence threshold (as the method's hyper-parameter) and apply Laplace-like regularization to prevent pair-weights of [0, 1] (absolute zero makes the log-likelihood -inf, so we regularize it to $[10^{-5}, 1 - 10^{-5}]$.) To avoid reinventing the wheel and adding extra methods to the field, as well as making more space in the main part of the paper we have switched to learnSPN and redone the analysis. No major conclusions have changed. We may add our structure learning method and its use in controlling free parameters as well as applying it to highly sparse data in the supplementary material.
>
> For the first two ``Other" points:  We apologize that this wasn't clear in our initial submission.  We have attempted to clarify the situation in our revision.  We test several different model configurations both with uniform and heterogeneous inputs. In both cases, we also provide specific input to pairs and quadruples to produce large beyond-pairwise correlations across many sets of neurons.
>
> For the last ``Other" point:  We apologize for this section.  We have rewritten it to improve clarity.

---

> > ### Comment · Reviewer_S4f9 · 2022-11-22
> > **Thanks for the clarification and additional baseline**
> >
> > Great to see that you included PLDS as another good baseline, and your model works better!
> >
> > Also thanks for clarifying the situation w.r.t. structure learning in SPNs. That's helpful to better assess the contributions of the paper.
> >
> > I am still a bit confused, though, about how you fit the models. Your revision hasn't made it clearer to me why you concatenate five trials and what "concatenate" means in this context. What are the 15 "blocks of population activity"? Could you clarify the dimensionality of one sample in your model, how many samples there are and how many total models you fit?

---

> > > ### Author Response · Authors · 2022-11-22
> > > **Thanks for your response!**
> > >
> > > We really appreciate reviewing our responses, and updating your score. We are happy that we addressed your concerns and apologize for the remaining lack of clarity of the fitting process. In this data set, there are 75 trials for each of the 4 directions ( 75 * 4 in total). We group/concatenate the time series for each of the 5 subsequent trials of which the same grating direction was shown to the mouse. For example, if the first 15 trials of drifting gratings have the following directions [0, 0, 90, 45, 90, 135, 0, 45, 0, 90, 0, 135, 0, 135, 45] the first group of trials for direction 0 consists of trials [1, 2, 7, 9, 11] (indexing from 1). Using 400 ms of data from each trial (from 400 to 800) with time bins of 20 ms gives us 20 data points for each trial. This means that our group of 5 trials consists of 20 * 5  = 100 data points. In other words, our data is a 100 * N binary matrix for this group (N is the number of neurons). We fit each model to this 100 * N matrix. Since there are 75 trials, we have 75/5 = 15 set of data-points/fit, meaning that each method (e.g. spn) is fit to fifteen 100*N data-set separately for each (mouse, area, direction). Overall, for each method (spn, rbm, etc), we fitted 60 = 15 * 4(directions) models for each pair of (mouse, area). For example, having data of 20 mice in area "VIsp" leads to 20 * 5 * 4 = 1200 fits of each method. For evaluation and statistical tests (fig 3 middle and right plots), the average Js-div in each of these 15 fits is calculated because all of these fits are from one mouse.
> > >
> > > Hyper parameters of each model for each of the 15 group/block were determined by cross validation within the data points of the group. For example, the number of hidden variables in the RBM was determined by 10-fold cross validation in each of the fits (fit on 90 data points, test on 10). This means that each of the 15 groups/block of a (mouse, area, direction) might have different numbers of hidden variables. Since PLDS is based on the whole trial, the number of latent variables is determined by leave-one-out cross validation on trials (with 5 trials in each block, this means fit to 4, test on 1).
> > >
> > > To further clarify, we used to have a "CV" fit (e.g. spn CV). In that one, the Js-div for each of the 15 group/block is calculated from samples generated from other 14 block models. Since, the difference between each of the 15 blocks could be because of changes in behavioral state of the animal, "data coming from the same distributing" assumption does not hold anymore and calling it cross-validation is not accurate (it is more like a mixture of other fits/models). Therefore, we removed that analysis (It was also consistent with our normal fit and not very informative).
> > >
> > > Fitting to movie clips followed the same procedure. The only difference was that because there were 60 movies instead of 75, each (mouse, clip, area) consisted of 12 fits instead of 15.

---

> > > > ### Comment · Reviewer_S4f9 · 2022-11-22
> > > > **Got it**
> > > >
> > > > Thanks, now I got it.
> > > >
> > > > However, that raises another issue I would strongly advise you to check: your samples within a trial are anything but independent, so I think you shouldn’t split 90/10 randomly but at least leave out entire trials for computing likelihoods, as you do for PLDS. Otherwise I would be worried that you might actually be overfitting quite substantially. If you look at the temporal cross correlogram of the neurons, they’re most likely not zero at 20 ms time lag.

---

> > > > > ### Author Response · Authors · 2022-12-07
> > > > > **Thanks for raising the issue! Our results/conclusions hold with the trial-CV**
> > > > >
> > > > > Thanks for warning about this issue which is especially important when compared to PLDS as at the very least, methods should be compared with the same conditions. Hyper parameters of SPN learning algorithm are "independence criterion/threshold", and the minimum number of data points required for the split/check independence which  make them not very sensitive to the parameter change when there are in a reasonable range. Due to this fact, lack of enough time to generate new results, and most importantly avoid over-fitting, we used the same set of hyper-parameters for all areas and mice, for each stimulus type after using (10 fold) cross validation for several examples in the revised manuscript for SPN (RBM's hyper parameters were tuned for each mouse-area-stimulus though). We did follow up with your raised concern and searched for optimal hyper-parameters of SPN by cross validation over trials (leave 1 entire trial out) in our experiments where we compared our model with PLDS (figure 5) (hence the delay in our response to your comment). Results of our new fits were close to previous one, and in fact in many mice/stimuli (slightly) better because of the tuning . More importantly, SPN's fits remained better than PLDS' in every mouse and stimulus. Unfortunately, we can not upload another revision at the moment with the updated figure 5, but the difference is not noticeable by eye. In the future versions of this work, our methodology and figure will be updated accordingly.

---

### Author Response · Authors · 2022-11-19
**Summary of the revision**

We wish to thank all the reviewers for their extensive and thought-provoking comments. We incorporated their feedback as much as possible, given the time limit. We believe the paper has improved significantly due to this feedback. The major additions to the paper are more synthetic data including different types of networks including heterogeneous networks, connected networks, and networks with various inputs (Section 4, simulation results, figure 2). We also added an analysis of 4 natural movie clips to our neural recording results (figure 4). Moreover, an additional baseline method, a Poisson latent dynamical system,  has been added to our empirical results (figure 5). Finally, we changed our structural learning approach to a slightly modified version of an existing method to improve the usability of SPNs for neural data and avoid adding extra methods to the field.

---

### Decision · Program_Chairs · 2023-01-20

**Decision:**

Accept: poster

**Justification For Why Not Higher Score:**

This is like a straightforward application of an existing method to a known (though long-standing) problem in neuroscience. Therefore, I don't see any specific reason it should be highlighted at ICLR.

**Justification For Why Not Lower Score:**

It is a straightforward accept - all reviewers recommended accept ([8,6,8,8]), and there are no-major concerns remaining.

**Metareview: Summary, Strengths And Weaknesses:**

This paper suggests using Sum-Product Networks (SPNs) for more accurate probabilistic modeling of neural populations. The results show improved accuracy compared to a few previous baselines. Reviewer had a few concerns, such as some missing baselines, but these concerns were sufficiently addressed, so eventually all reviewers recommend accepting the paper. I recommend the authors to finish the comparison with all the baselines mentioned in the review (or explain why some baselines were omitted).

**Note From Pc:**

if the above contains the word "oral" or "spotlight" please see: "oral" presentation means -> notable-top-5% and "spotlight" means -> notable-top-25%. As stated in our emails, we are disassociating presentation type from AC recommendations